# The Tyrosine Phosphatase SHP2: A New Target for Insulin Resistance?

**DOI:** 10.3390/biomedicines10092139

**Published:** 2022-08-31

**Authors:** Céline Saint-Laurent, Laurène Mazeyrie, Mylène Tajan, Romain Paccoud, Isabelle Castan-Laurell, Philippe Valet, Thomas Edouard, Jean-Philippe Pradère, Cédric Dray, Armelle Yart

**Affiliations:** 1RESTORE Research Center, Université de Toulouse, Institut National de la Santé Et de la Recherche Médicale 1301, Centre National de la Recherche Scientifique 5070, Etablissement Français du Sang, Ecole Nationale Vétérinaire de Toulouse, 31100 Toulouse, France; 2Endocrine, Bone Diseases and Genetic Unit, Reference Center for Rare Diseases of Calcium and Phosphate Metabolism, ERN BOND, OSCAR Network, Pediatric Research Department, Children’s Hospital, Toulouse University Hospital, 31059 Toulouse, France

**Keywords:** SH2 containing protein tyrosine phosphatase 2, Noonan syndrome, insulin resistance, macrophage, inflammation

## Abstract

The SH2 containing protein tyrosine phosphatase 2(SHP2) plays essential roles in fundamental signaling pathways, conferring on it versatile physiological functions during development and in homeostasis maintenance, and leading to major pathological outcomes when dysregulated. Many studies have documented that SHP2 modulation disrupted glucose homeostasis, pointing out a relationship between its dysfunction and insulin resistance, and the therapeutic potential of its targeting. While studies from cellular or tissue-specific models concluded on both pros-and-cons effects of SHP2 on insulin resistance, recent data from integrated systems argued for an insulin resistance promoting role for SHP2, and therefore a therapeutic benefit of its inhibition. In this review, we will summarize the general knowledge of SHP2’s molecular, cellular, and physiological functions, explaining the pathophysiological impact of its dysfunctions, then discuss its protective or promoting roles in insulin resistance as well as the potency and limitations of its pharmacological modulation.

## 1. The Tyrosine Phosphatase SHP2

The SH2 domain-containing tyrosine phosphatase 2 (SHP2) is a ubiquitous protein tyrosine phosphatase (PTP) with conserved structure and function from invertebrates to humans. Thanks to its Src Homology 2 (SH2) domains, SHP2 is recruited and activated by phosphotyrosines borne by membrane receptors or anchorage proteins, and then dephosphorylates other phosphotyrosine residues, thereby regulating protein/protein interactions, subcellular localization of proteins, phosphorylation-dependent activity of enzymes, etc. In particular, SHP2 has been proved to be a positive regulator of the RAS/mitogen-activated protein kinase (MAPK) signaling cascade, and a negative regulator of the phosphoinositide-3 Kinase (PI3K)/AKT pathway. SHP2 is mobilized downstream from many membrane receptors activated by growth factors, hormones, cytokines, or adhesion molecules, which confers fundamental roles on SHP2 in key cellular processes (proliferation, differentiation, apoptosis…). As a result, SHP2 has pleiotropic functions during development and homeostasis maintenance, and is associated, when dysregulated, with different pathologies ranging from genetic diseases such as Noonan syndrome (NS), to cancers.

In this chapter, we will briefly review the knowledge about SHP2 function(s) at the molecular and cellular levels, then outline the consequences of its invalidation in various tissues and organs to highlight its roles in fundamental processes such as stemness, regeneration, organogenesis, or energy metabolism. The last part will be devoted to the description of diseases associated with SHP2 dysregulation.

### 1.1. Structure, Function and Regulation

SHP2 is a ubiquitous non-receptor protein tyrosine phosphatase (PTP), which forms, together with SHP1, its close hematopoietic-restricted relative, the SH2-domain-containing, non-receptor PTPs subgroup. SHP2/Shp2 proteins are highly conserved along the Vertebrate subphylum (>80% identity, >95% among mammals), and orthologs are found in non-vertebrates, notably insects (Corskrew/CSW in *D. melanogaster*) and roundworms (PTP-2 in *C. elegans*). SHP2 is composed of two N-terminal SH2 domains (N-SH2 and C-SH2) able to interact with phosphotyrosine-containing molecules, a C-terminal domain containing phosphorylation sites and a protein/protein interaction motif with regulatory functions, and a central PTP catalytic domain carrying a canonical PTP signature motif (C(X)_5_R), that dephosphorylates specific phosphotyrosines [1,2]. Recent biochemical studies, as well as machine learning approaches, directed towards the identification of SHP2′ substrate specificity, revealed an enrichment of acidic residues upstream from the phosphotyrosine [3,4].

A sophisticated molecular switch, relying on complex intra and extramolecular interactions, controls SHP2 activation. At basal state, its N-SH2 domain caps the enzymatic pocket, folding the protein into an auto-inhibited, closed conformation that prevents the access of SHP2’s substrates to the enzymatic cleft. The engagement of the SH2 domains by phosphotyrosine-containing signaling partners, or in some cases by SHP2’s own phosphotyrosines (Y542 and Y580), releases these inhibitory interactions and reshapes the phosphatase into an open, active, conformation that allows the binding of phosphotyrosine-containing substrates to the enzymatic site and their dephosphorylation [2]. Between the closed and open conformation, a partially open semiactive state has recently been described, that mediates intermediate signaling tone [5]. Interestingly, it has been also demonstrated that these conformational changes favor a liquid–liquid phase separation (LLPS) behavior that concentrates specific biochemical reactions [6], but also promotes an eased access to the active conformation [7]. As we will see later, these properties have been used to develop specific allosteric inhibitors. In addition, SHP2’s activity is negatively regulated by several post-translational modifications and/or protein/protein interactions, including inhibitory serine/threonine phosphorylations (Protein Kinase A-dependent T73/S189 phosphorylation) [8], competitive binding of Growth factor receptor-bound protein 2 (GRB2) to SHP2′ C-terminal phosphotyrosines [9], Yes-Associated Protein/ Transcriptional coactivator with PDZ-binding motif (YAP/TAZ)-dependent cytoplasmic sequestration [10], or reactive oxygen species (ROS)-mediated oxidation of the catalytical cysteine, involving, for example, Nicotinamide adenine di-nucleotide phosphate (NADPH) oxidase (NOX) [11,12,13].

As a general scheme, the dephosphorylation of phosphotyrosine residues on specific substrates modifies their conformations or their binding properties, thereby modifying enzymatic activity, recruitment to signaling complexes, and/or subcellular localization. Beside its enzymatic activity, there is evidence of PTP-independent docking functions for SHP2. Thus, the main role of SHP2, like other PTPs, is to regulate various signal transduction pathways that coordinate cell adaptation to changes in its environment, that are sensed by membrane receptors (e.g., receptor tyrosine kinases, RTK), or to intrinsic modifications. Indeed, critical steps of these signaling cascades are triggered by kinase/phosphatase-mediated phosphorylation/dephosphorylation events, the regulation of which must be fine-tuned. However, in contrast to most of the PTPs that are seen as counteractors of tyrosine kinase-mediated activating cues, SHP2 has a rather positive function in cell signaling. Its best-known role is indeed to contribute, downstream from many growth factors, cytokines, and hormone receptors, to the activation of the RAS/mitogen-activated protein kinases (MAPK) extracellular-regulated kinases 1/2 (ERK1/2) pathway, an essential and versatile route that regulates many fundamental cell functions, including proliferation, apoptosis, differentiation, motility, morphology, or metabolism, which notably confers a proto-oncogene status on SHP2 [14]. Depending on the agonist and the cell/tissue, different mechanisms have been described to explain SHP2-dependent RAS/MAPK activation, including GRB2/Son Of Sevenless (SOS) recruitment, SRC activation, release of SPRY-mediated RAS inhibition, RAS GTPase Activating Protein (RASGAP) inhibition, or dephosphorylation of RAS inhibitory, GTPase-stimulating, Y32 residue [15,16,17].

Besides RAS/MAPK, SHP2 has also been described as a negative regulator of the phosphoinositide 3-kinase (PI3K), through its ability to dephosphorylate the p85 regulatory subunit-binding sites (Y-X-X-M motif) on docking proteins such as GAB1 or IRS1 [18]. Indeed, some of them are genuine SHP2 substrates, with acidic residues preceding the phosphotyrosine (e.g., HTDDGpYMPM for Y612 on IRS1, SEENpYVPM for Y589 on GAB1). We will describe the importance of this mechanism in the regulation of insulin signaling (see next section). In addition, depending on the agonist and cell type, SHP2 can regulate many other signaling cascades, such as Janus Kinases/Signal Transducers and Activators of Transcription (JAK/STAT), nuclear factor-kappa B (NFκB), Programmed death-1/PD-ligand 1 (PD-1/PD-L1), various kinases (Adenosine Monophosphate-activated protein kinase AMPK, c-Jun N-terminal Kinase JNK, Target Of Rapamycin TOR), or cytoskeleton proteins (Rho-associated protein kinase ROCK, Paxillin PXN) (reviewed in [2]). Non-signaling functions have also been described for SHP2: SHP2 can translocate to the mitochondria following sepsis in cardiomyocytes or upon NOD-like receptor family, pyrin domain containing 3 (NLRP3) inflammasome activation in macrophages, where it dephosphorylates adenine nucleotide translocase 1 (ANT1) or components of the Oxidative Phosphorylation (OXPHOS) complex [19,20,21], thereby regulating mitochondria activity. Nuclear import of SHP2/STAT5 complex has been reported, notably upon prolactin stimulation [22]. In the nucleus, SHP2 plays roles in transcriptional regulation as well as in modulating telomerase activity [23,24]. SHP2 also localizes to the nucleus thanks to its interaction with the YAP/TAZ proteins, where it dephosphorylates the protein parafibromin, shifting it to its pro-oncogenic, WNT/CTNNB1 (β-catenin)-promoting activity [10]. Moreover, SHP2 physically interacts with ERα and regulates estrogen-dependent transcriptional programs [25,26]. Interestingly, unbiased approaches using phosphoproteomic strategies have recently broadened the landscape of SHP2-regulated phosphorylation events and algorithms aiming at predicting substrate specificity, combined with deep learning approaches, have identified potential new substrates, such as occludin or Phospholipase Cγ2 (PLCγ2) [4,27,28,29,30].

### 1.2. Physiological Roles of SHP2 during Development and Homeostasis, and Pathological Consequences of Its Dysfunction

The fundamental role of SHP2 in regulating canonical signaling pathways, its ubiquitous expression, and its mobilization downstream from many, if not all, growth factors, cytokines, and hormones receptors make SHP2 a key actor in numerous developmental and homeostatic processes. Indeed, SHP2 invalidation consistently impairs organism development in various animal models, which notably relates to very early functions of SHP2 in stem cell and progenitor self-renewal and differentiation, with consequences on morphogenesis and embryo patterning [1]. These effects have often been linked to RAS/MAPK ERK1/2 hypoactivation in response to different stimuli, including Leukemia Inhibitory Factor (LIF), bone morphogenetic proteins (BMP), Stem Cell Factor (SCF), or Fibroblast Growth Factor (FGF), but other signaling defects have been impleaded [31]. Models of tissue-specific invalidation have extended this role of SHP2 to the development and specification of many tissues and organs, covering the whole of organ systems in mammals (integumentary, musculoskeletal, hematopoietic, nervous, endocrine, metabolic, circulatory, respiratory, digestive, urinary, and reproductive) (for review see [1]).

Given its canonical role in mitogenic response and in the regulation of proliferation and survival of many cells, SHP2’s oncogenic properties need no further proof [32]. Indeed, the SHP2-encoding gene, protein tyrosine phosphatase non-receptor type 11 (*PTPN11),* has been defined as a bona fide proto-oncogene and is involved in tumorigenesis, being overexpressed in various cancers or relaying proliferation signals from upstream oncogenes, such as Epidermal Growth Factor Receptor (EGFR), KIT, or Fms-like tyrosine kinase 3 (FLT3) mutants. Moreover, somatic mutations, resulting in SHP2 hyperactivation, are found in a wide range of cancers, including juvenile myelomonocytic leukemias (JMML, MIM #607785) and other myeloid neoplasms [2,33,34]. Thus, specific inhibitors have recently emerged as potent therapeutic strategies to combat cancer or overcome cancer resistance, opening promising avenues in the field of cancer therapy [2]. Intriguingly, a tumor suppressor activity has been also described for SHP2, notably in liver cancer, cartilage tumors, or esophageal carcinoma [35,36], which may be linked to the induction of an immunosuppressive environment [37,38]. Consistent with its key functions in organism patterning, germline mutations of *PTPN11* have also been described as the causal factor of several rare developmental, autosomal dominant, disorders. Noonan syndrome (NS, MIM #163950) and NS with multiple lentigines (NS-ML, MIM 151100) belong to one of the largest groups of multiple anomalies syndromes, which share a wide spectrum of congenital defects including cardiopathies, growth delay, cranio-facial abnormalities, hematological defects, mental retardation, and tumor predisposition. With related diseases (e.g., Costello syndrome, cardio-facio-cutaneous syndrome), they form the family of RASopathies, named after the causal mutations hitting genes encoding intermediates of the RAS/MAPK pathway, and mostly resulting in its hyperactivation [39]. In addition, loss-of-function mutations (frameshift, nonsense, or splice site mutations) have been allocated to metachondromatosis (MC, MIM #156250), a very rare dominant condition in which patients develop multiple bone benign tumors (enchondromas and osteochondromas) following loss-of-heterozygosity events [40].

As a signaling intermediate of numerous hormones and cytokines, SHP2 also participates in homeostasis maintenance. To cite only a few, SHP2 modulates energy balance through central leptin-dependent food intake and energy expenditure regulation [25,41,42] and peripheral control of energetic substrates (glucose and lipid) storage/handling in metabolic tissues [43,44,45,46,47]. In particular, it plays a key role in insulin-mediated control of glucose metabolism, which will be detailed in the next section. SHP2 also regulates organism growth, via systemic effects on the growth hormone (GH)/insulin-like growth factor 1 (IGF-1) axis and local effects on bone homeostasis [48,49]. In addition, many studies have highlighted important roles for SHP2 in innate and adaptative immune response, as it contributes to myelopoiesis and macrophage polarization, T cell selection or activation, or B cell expansion [50,51,52].

## 2. SHP2, in Insulin Resistance: A Fragmented and Confusing View from Cellular and Tissue Specific Models

As for many other RTKs, SHP2 is a central hub in insulin signaling. While it plays a positive role in the mitogenic function of insulin, it has been proved to negatively regulate the metabolic arm of insulin signaling through complex mechanisms (Insulin receptor substrate-1 [IRS1] dephosphorylation, PI3K regulation…). The implementation of genetically engineered models has highlighted a complex interplay between SHP2 and glucose metabolism regulation, with conflicting results depending on the targeted tissue. Indeed, SHP2 invalidation in skeletal muscle or central nervous system results in insulin resistance but is protective against insulin resistance when targeted in the hepatocyte or in the macrophage. Adding further complexity, SHP2 disruption in pancreatic β cells impaired insulin biosynthesis and secretion.

### 2.1. General Considerations about Insulin Signaling and Insulin Resistance

Maintenance of glucose homeostasis, operating through a fine-tuned regulation of glucose absorption, production, storage, and use, is essential for energy supply and thus for vital functions. Produced and secreted by the pancreas when glycemia rises, insulin is a major hypoglycemic hormone, that triggers glucose uptake by white adipose tissue (WAT) and skeletal muscle, and stimulates its use (glycolysis) or its storage as macromolecules such as glycogen (glycogenogenesis in liver and muscle) and triglycerides (de novo lipogenesis in WAT and liver). It also inhibits hepatic glucose production (gluconeogenesis and glycogenolysis). Tacked on to that are indirect regulatory mechanisms, involving cytokine-dependent inter-organ dialogues or inhibition of glucagon secretion, as well as central regulation. Beyond its key hypoglycemic role, insulin has also anabolic long-term properties, acting on organismal growth and lifespan [53,54,55]. Insulin mediates its effects at the cellular level by stimulating its cognate cell surface receptor tyrosine kinase (InsR), which once phosphorylated initiates several signaling pathways, directly or via the recruitment of docking proteins such as IRS proteins. Recruitment and activation of class I PI3K, through the binding of the p85 regulatory subunit to the Y-X-X-M motif borne by IRS, mediates most of the hypoglycemic effects of insulin. Indeed, AKT activation triggers GLUT4-dependent glucose uptake, promotes glycogenogenesis by phosphorylating Glycogen Synthase Kinase 3 (GSK3), and inhibits the neoglucogenesis transcription program in a Forkhead Box O1 (FOXO1)-dependent manner [56]. The RAS/MAPK pathway is also directly mobilized by InsR to mediate its mitogenic effect and to downregulate the PI3K/AKT arm through ERK1/2-dependent phosphorylations of IRS on specific residues, that inhibit IRS/P85 binding. Insulin signaling is finally reset by Protein Tyrosine Phosphatase 1B (PTP1B)-dependent dephosphorylation of activating tyrosines borne by InsR and IRS1, as well as InsR internalization and recycling or degradation [56].

Insulin resistance is defined as a loss of response of target tissues to insulin stimulation, which critically affects the control of glycemia, leading to hyperglycemia and contributing to type 2 diabetes (T2D), as well as lipid metabolism, notably through the loss of insulin anti-lipolytic action. With half a billion affected patients, T2D is a very frequent metabolic disease intimately linked to the pandemic of obesity and metabolic syndrome. Many processes have been described to drive insulin resistance, among which excessive InsR stimulation in a context of hyperinsulinemia, leading to increased InsR recycling and subsequent reduced InsR molecules at the plasma membrane, and spatiotemporal inhibition of insulin-evoked signaling, through dephosphorylation of activating tyrosine residues, phosphorylation of inhibitory serine/threonine residues, or dephosphorylation of PI3K lipid products. Besides intrinsic defects, insulin resistance can be triggered by indirect mechanisms, including lipotoxicity and inflammation, which are often interrelated. Lipotoxicity results from increased serum free fatty acids (FFA) levels and ectopic lipid deposits, which occurs when the storage capacity of adipocytes is overwhelmed (e.g., during obesity) or impaired (e.g., in lipoatrophic or lipodystrophic diseases), and is subjected to a vicious cycle through loss of insulin’s antilipolytic effect in insulin resistant WAT. Increased diacyglycerol (DAG) levels within lipid droplets, ceramides, and circulating free fatty acids (FFA) have been shown to activate serine/threonine kinases, such as novel Protein Kinases C or JNK (see [56] for extensive review), which inhibit proximal insulin signaling by phosphorylating InsR or IRS1. Although the molecular mechanisms are not fully understood, mitochondria and endoplasmic reticulum (ER) dysfunctions, metabolic dysregulation, and impaired nutrient sensing also contribute to lipid-mediated insulin resistance in different metabolic tissues. Additionally, low-grade inflammation of metabolic tissues, combining increased proinflammatory cytokines (tumor necrosis factor TNFalpha, Interleukin 1 IL1beta, IL6…) levels and activation of inflammatory infiltrated and resident macrophages and other immune cells, has emerged as a central mechanism of insulin resistance. Many triggers of inflammation in insulin resistance-associated diseases, especially obesity, have been documented, including non-exhaustively, fatty acids, adipocytes/hepatocytes death, mitochondria dysfunction and loss of metabolic flexibility, or ER stress (for review, see [57]). Then, the mechanisms connecting inflammation to insulin resistance are also multifactorial, involving different cytokine-evoked signaling pathways, such as IkappaB kinase (IKK)/NF-κB, JNK/p38MAPK, PKC-dependent serine phosphorylation of IRS1 or InsR, or suppressor of cytokine signaling (SOCS)-mediated insulin signaling downregulation [58] (Figure 1).

### 2.2. A Direct and Dual Role of SHP2 in Insulin Signaling

InsR being a prototypical RTK and recruiting canonical signaling pathways, the contribution of SHP2 in the regulation of insulin-dependent response has been extensively studied using complementary approaches in vitro. As observed upon stimulation of other RTK, SHP2 inhibits PI3K/AKT in response to insulin in different cell types, notably by dephosphorylating the p85 binding sites (Y-X-X-M) borne by IRS1 [18,59,60,61]. In accordance with these findings, insulin signaling is enhanced in tissues from mice with SHP2 loss of expression/activity, although it does not always translate into improved systemic insulin sensitivity (see Section 2.3) [44,46,60]. However, consistent with a role of SHP2 in promoting cellular insulin resistance by inhibiting insulin-evoked PI3K signaling, high sugar exposure-induced insulin-resistant hepatocytes overexpress SHP2 and display reduced PI3K/AKT signaling. This effect translated into reduced glucose consumption, and SHP2 inhibition or loss of expression rescued the insulin resistance phenotype [62]. At least in hepatocytes, SHP2-dependent inhibition of the PI3K-dependent arm of insulin signaling could be triggered by binding to Afadin 6 (AF6), a polarity protein that contributes to insulin resistance [63]. Moreover, in insulin-resistant cells, SHP2 hyperactivates the RAS/MAPK cascade, which downregulates the PI3K/AKT pathway by phosphorylating serine residues immediately following the Y-X-X-M motif. SHP2-mediated RAS/MAPK activation has also been shown to mediate InsR endocytosis [64,65]. Therefore, SHP2 has rather a negative role in insulin-evoked PI3K signaling in vitro that contributes to insulin resistance.

### 2.3. A protective or Causal Role of SHP2 in Insulin Resistance In Vivo?

Establishing an in vivo role for SHP2 in glucose metabolism has relied on animal studies. In non-vertebrate models, impairment of insulin signaling upon global invalidation of SHP2 orthologs (Csw in fly, Ptp-2 in worm) has been reported, but assigned to global lifespan regulation, without assessing glucose metabolism [53,55]. In mice, *Ptpn11* total knock out is embryonic lethal, precluding the evaluation of SHP2 global role in glucose metabolism, but heterozygous animals did not show any obvious phenotype, at least in the absence of metabolic challenge [66]. In contradiction with the in vitro findings, transgenic model overexpressing an inactive mutant of SHP2 displayed an insulin resistance phenotype associated with PI3K/AKT downregulation [67]. The advent of genetically engineered animal models, aiming at suppressing SHP2 expression or expressing hyperactive mutants in specific tissues or organs, allowed documenting the metabolic impact of its targeted dysfunction and further supported a protective role for SHP2 in insulin resistance. Indeed, several models of neuronal inactivation of SHP2 display insulin resistance, while knocking in a hyperactive mutant of SHP2 in forebrain neurons protects from glucose intolerance and insulin resistance. However, these phenotypes correlated with obesity/hyperphagia and protection against high-fat-diet (HFD)-induced obesity, respectively, in favor of an indirect mechanism driven by central, leptin-dependent signals [25,41,68]. Beta cell-specific disruption of the SHP2 encoding gene highlighted a key role for SHP2 in insulin biosynthesis and glucose-stimulated insulin secretion, correlated with reduced expression of insulin-encoding genes (*Ins1*, *Ins2*) and glucose sensor glucose transporter isoform 2 (GLUT2). As a consequence, SHP2^panc−/−^ mice developed glucose intolerance with age, but their phenotype upon metabolic challenge was not documented [45]. Muscle-specific SHP2 invalidation resulted in reduced insulin-evoked glucose uptake in skeletal muscle and systemic insulin and glucose intolerance [46]. Depending on the construction used, adipocyte-specific models of SHP2 invalidation develop variable phenotypes ranging from unaltered adipose tissue function to adipogenesis inhibition and severe lipoatrophy, which did not allow to conclude about its impact on glycemia control [43,69]. However, in a recent study using a model of sleeve gastrectomy in rats, Qi et al. documented a negative role for SHP2 in adipogenesis, which could translate into improved insulin sensitivity in this specific condition [70].

Other studies supported a promoting role for SHP2 in insulin resistance. Indeed, hepatocyte-specific *Ptpn11* knock-out mice displayed better glucose tolerance and insulin sensitivity, which translated into protection from HFD-induced insulin resistance and T2D. This phenotype was consistently associated with a hyperactivation of insulin-evoked PI3K/AKT and was also induced by acute hepatic deletion of SHP2 through adenoviral Cre approaches, arguing for a direct effect [44,47]. Conversely, overexpression of SHP2 using adenovectors resulted in glucose intolerance, insulin resistance, and impaired insulin signaling [63]. Interestingly, a recent study revealed that targeted disruption of SHP2 in macrophages had a protective effect towards HFD-induced insulin resistance, which was assigned to its role in repressing IL18 production [71] (Figure 1).

## 3. SHP2 in Insulin Resistance: Lessons from Integrated Models

While the growing number of preclinical models with sometimes contradictory phenotypes is somehow puzzling, there is convincing evidence for a role of SHP2 in insulin resistance. First, *PTPN11* has been identified as one of the potential candidates for metabolic syndrome in humans. Moreover, patients with Noonan syndrome, carrying a hyperactive form of SHP2, have been recently shown to display glucose intolerance. This phenotype has been ascribed to inflammation-driven insulin resistance in an NS animal model carrying an activating mutation of SHP2, while, in contrast, a mouse model expressing an inactivating mutation of SHP2 has increased glucose tolerance and is protected from HFD-induced insulin resistance. In addition, SHP2 overexpression has been detected in metabolic tissues of several animal models of obesity/diabetes, and in blood cells from diabetic patients, although the cause of this dysregulation is still misunderstood. Consistent with a rather global role of SHP2 in promoting insulin resistance, recent studies have taken advantage of the last generation of SHP2 inhibitors to document that pharmacological inhibition of SHP2 alleviates insulin resistance in obese/diabetic mice. In this section, we will review the recent advances regarding the consequences of systemic SHP2 dysregulation on glucose metabolism and discuss how this knowledge helped improve our global understanding of insulin resistance.

### 3.1. Genetic Diseases, Susceptibility Gene and SHP2 Dysregulation in Obesity/Diabetes

As mentioned above, *PTPN11* germline mutations are associated with genetic diseases. Although these diseases are not characterized by an obvious or severe metabolic phenotype, they represent a unique situation with constitutive and systemic perturbation of SHP2’s function, providing a potent model to assess SHP2’s global role in glucose regulation and the long-term outcome of its systemic perturbation. Interestingly, several recent clinical studies have reported an unsuspected metabolic profile in patients with NS, notably those carrying a *PTPN11* mutation. NS patients, whether children or adults, actually displayed low body mass index/adiposity and hypocholesterolemia, but glucose intolerance or insulin resistance, most often associated with lower-than-average or normal basal glycemia and insulinemia [72,73,74]. Similarly, a mouse model of NS (SHP2 ^D61G/+^) concomitantly developed insulin resistance and reduced adiposity, which was neither associated with a lipodystrophic or lipoatrophic condition, nor with ectopic lipid deposits, arguing against lipotoxicity as the cause of insulin resistance [74]. Supporting this view, this phenotype was associated with a constitutive inflammation of the metabolic tissues and further explorations pinpointed that hyperactive SHP2 modified resident macrophage homeostasis, notably in the liver (Kupffer cells, KC), and triggered monocyte infiltration [74]. Importantly, restricted SHP2 hyperactivation in the hematopoietic compartment, by the mean of bone marrow transplantation, was sufficient to promote insulin resistance, while repopulation of NS animals with wildtype bone marrow cells, as well as macrophage depletion, alleviated their insulin resistance. Consistent with the fact that SHP2 hyperactivation in the myeloid population was sufficient to promote insulin resistance in NS mice, its macrophage-targeted disruption protected HFD-fed mice against insulin resistance [71]. Therefore, in the context of whole-body SHP2 hyperactivation, the insulin resistance-promoting effect of SHP2 in the macrophage and/or in the hepatocytes, seems to outshine its protective role in other tissues.

Mirroring these findings, a mouse model of NS-ML, expressing a hypoactivating mutation of SHP2 (SHP2 ^T468M/+^), displayed insulin hypersensitivity and protection to HFD-induced insulin resistance, although it shares with NS the reduced adiposity. A direct signaling role of NS-ML-causing SHP2 mutants on insulin-evoked PI3K activation was proposed since it could be observed in both insulin-sensitive tissues and cultured cells, and be recapitulated in cultured cells [60]. However, whether this protection against insulin resistance also goes through an anti-inflammatory effect of SHP2 hypoactivation, as suggested in models [50,71,75,76,77,78,79,80], has not been assessed in the NS-ML mouse model.

Beyond rare diseases, *PTPN11* has been reported as one of the 25 susceptibility genes for metabolic syndrome in humans, a common condition associated with cardiovascular diseases, metabolic disorders and insulin resistance, and inflammation [81]. In addition to *PTPN11* genetic variants, SHP2 overexpression has also been associated with glucose metabolism alteration and insulin resistance. Indeed, SHP2 expression, and possibly its activity, is increased in the liver, muscle, and white adipose tissue of obese/diabetic animals or HFD-fed mice [47,69,70,82,83]. While the molecular mechanisms controlling SHP2 expression are still misunderstood, it seems sensitive to the nutritional state, being repressed in the liver upon fasting and in the WAT upon leptin treatment, and is also induced in vitro in high-sugar exposure-induced insulin-resistant cultured hepatocytes [47,62,70]. Moreover, consistent with its specific role in macrophages, palmitic acid treatment stimulated SHP2 phosphorylation in murine bone marrow-derived macrophages (BMDM) and increased SHP2 activity has been reported in peripheral blood mononuclear cells (PBMC) from patients with T2D [71].

### 3.2. SHP2 Targeting in Obesity/Diabetes

Based on the “diabesity” phenotype of mouse models of brain-specific *PTPN11* disruption and SHP2’s functions in leptin signaling and energy balance regulation, it has been initially proposed that SHP2 activators could represent interesting therapeutic approaches to alleviate obesity-associated insulin resistance [41,42,68,84]. However, in accordance with accumulating evidence for a role of SHP2 in promoting insulin resistance, the therapeutic potential of its inhibition has also emerged. While SHP2, like other PTPs, has long been considered undruggable because of the structure and conservation of its catalytical domain, preventing sufficient selectivity and specificity, a better understanding of its structure and regulatory mechanisms provided new options for drug development, notably for allosteric inhibitors and proteolysis-targeting chimera (PROTAC)-based degraders. This new generation of molecules, which display high selectivity and specificity, good tolerance, and bioavailability, has considerably enriched the pharmacological toolbox to study SHP2 pathophysiological functions, highlighting the therapeutic potential for inflammatory diseases and cancer in preclinical models. Thus, several of them have already moved on to clinical trials (e.g., BBP-398, NCT05375084; TNO155, NCT04000529) in the field of cancer treatment (for recent reviews, see [2,32,50]). Interestingly, several studies recently reported a positive outcome of SHP2 inhibition on insulin resistance. Indeed, a 2-week treatment of HFD-fed obese mice with SHP099, an allosteric, bioavailable SHP2 inhibitor, improved their glucose tolerance and insulin sensitivity, an effect that came along with an improvement of hepatic steatosis. Noticeably, this effect was similar and even stronger than widely used anti-diabetics (e.g., metformin, rosiglitazone) and was specifically associated with an improvement of the inflammatory status, a reduction in inflammatory markers and macrophage infiltrate in metabolic tissues, and a decrease in hepatic KC markers [74]. Moreover, SHP2 inhibition correlated with an increase in IL18 plasma levels, which contributed to alleviate insulin resistance [71]. Interestingly, while brain-targeted SHP2 gene invalidation translated into severe weight gain, systemic SHP2 inhibition did not worsen the obesity phenotype of the HFD-fed mice, probably because leptin signaling was already blunted [41,42,68,84]. Since SHP099 effectively crosses the blood–brain barrier [85], evaluation of a possible weight-gaining effect will be required on lean, leptin-sensitive animals (Table 1).

## 4. Conclusions and Future Directions

To sum up, SHP2 shows versatile roles in insulin resistance, ranging from signaling mechanisms to systemic regulations, with different, even opposite, effects depending on the considered cell, tissue, or context. However, studies on integrated preclinical models as well as in humans argue in favor of an insulin resistance-promoting role for SHP2, pointing out the therapeutic potential of its inhibition. While numerous determinants of insulin resistance have been identified so far, the therapeutic potential of their targeting often turned out to be limited, due to compensatory mechanisms or off-target effects. In contrast, the fact that SHP2 hyperactivation directly promotes insulin resistance in patients with NS highlights its pathophysiological power in an integrated system that is directly relevant to human disease, and promising preclinical results have been obtained with the recently developed SHP2 inhibitors. Moreover, by targeting both insulin signaling and systemic mechanisms, SHP2 inhibition seems to offer a wider potential compared to existing anti-diabetics or other candidates acting only on one process (e.g., PTP1B inhibitors to reactivate insulin signaling) [86]. Yet, many unknowns remain regarding the respective contribution of the different tissues and their interplays, and the cellular and molecular mechanisms that trigger SHP2 dysregulation in the context of obesity/diabetes. Answering these questions will certainly unlock major pitfalls related to SHP2 pleiotropic roles, that could preclude its systemic inhibition as a therapeutic approach but also open new areas for therapeutic intervention. Indeed, specifically inhibiting SHP2 in cells or tissues that actually overexpress/overactivate it by the mean of directed drug delivery could be more tolerated, and targeting the processes that promote SHP2 dysregulation and/or the cells in which its overactivation occurs may represent an alternative.

## Figures and Tables

**Figure 1 biomedicines-10-02139-f001:**
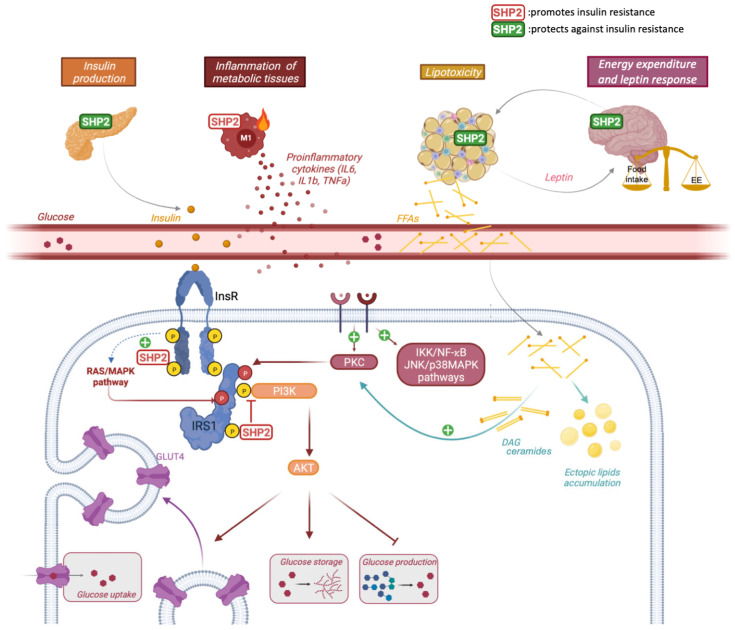
SH2 containing protein tyrosine phosphatase (SHP2) in insulin signaling: At the cellular level, SHP2 plays a negative role in insulin metabolic pathway by dephosphorylating phosphoinositide 3-kinase (PI3K)-binding sites (in yellow) and through RAS/Mitogen Activated Protein Kinase (MAPK)-dependent inhibitory phosphorylation (in red). This results in the downregulation of insulin-dependent glucose uptake and storage, and the increase of glucose production, notably in the liver, all contributing to hyperglycemia. In addition, SHP2 promotes a proinflammatory M1 state in macrophages, which triggers inflammation-dependent insulin resistance. On the other side, SHP2 has protective effects against insulin resistance by (i) promoting insulin biosynthesis in pancreas, (ii) regulating the development and function of adipose tissue, and (iii) stimulating central, leptin-dependent, energy expenditure. DAG: diacylglycerol, FFA: free fatty acid, InsR: insulin receptor. Created with BioRender.com accessed on 30 June 2022.

**Table 1 biomedicines-10-02139-t001:** Impact of SHP2 dysregulation on insulin resistance.

Targeted Organism/Tissue/Cell	Approach	Impact on Glucose Metabolism and Major Associated Phenotype	References
Cultured hepatocytes	SHP2 knockdown or inhibition	Improved insulin signaling	[59,63,64]
*C. elegans*	*Ptp2* ^−/−^	Modulation of insulin signaling Increased lifespan	[53]
Drosophila	Csw^−/−^	Modulation of insulin signaling Increased lifespan	[55]
Mouse/ubiquitous	*Ptpn11*−/− *Ptpn11*+/−	Undetermined (embryonic lethality) No obvious phenotype	[66]
Mouse/ubiquitous	*Ptpn11* ^T468M/+^	Insulin hypersensitivity Improved glucose tolerance Resistance to HFD-induced insulin resistance	[61]
Mouse/ubiquitous	*Ptpn11* ^D61G/+^	Insulin resistance Glucose intolerance Inflammation	[74]
Mouse/transgenic (liver, muscle, adipose tissue)	*SHP2*∆*PTP*	Insulin resistance	[67]
HFD-fed mouse/systemic	*SHP099 treatment*	Improved glucose tolerance	[71,74]
Mouse/muscle specific	*Ptpn11^fl/fl^* × MHC-cre or MCK-cre	Insulin resistance Glucose intolerance Altered myofibers (number and size) Dilated cardiomyopathy	[46]
Mouse/neuron specific	*Ptpn11^fl/fl^* × CRE3	Insulin resistance Diabetes Obesity Hyperphagia Leptin resistance Nephropathy	[41,68]
Mouse/transgenic (forebrain neuron, CamKII-driven)	*SHP2^D61A^*	Improved insulin sensitivity Improved glucose homeostasis	[25]
Mouse/POMC neuron specific	*Ptpn11^fl/fl^* × POMC-cre	Altered glucose metabolism Obesity Leptin resistance Reduced energy expenditure	[41,84]
Mouse/liver specific	*Ptpn11^fl/fl^* × Alb-cre	Improved insulin sensitivity Improved glucose tolerance Resistance to obesity Increased energy expenditure	[44,47]
Mouse/adipose tissue specific	*Ptpn11^fl/fl^* × Adipoq-cre	No phenotype	[69]
	*Ptpn11^fl/fl^* × aP2-cre	Not assessed Severe lipodystrophy Altered adipogenesis Premature death	[43]
Mouse/pancreas specific	*Ptpn11^fl/fl^* × Pdx1-cre	Glucose intolerance Reduced insulin secretion	[45]
Mouse/macrophage specific	*Ptpn11^fl/fl^* × Lyz2-cre	Resistance to HFD-induced insulin resistance	[71]
Human/ubiquitous	NS-*Ptpn11*	Glucose intolerance Noonan syndrome	[72,73,74]

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
