# Peer review of "The Tyrosine Phosphatase SHP2: A New Target for Insulin Resistance?"

_biomedicines, 2022, doi:10.3390/biomedicines10092139_

Round 1
Reviewer 1 Report
This article seems to be perfectly fine to me. However, I suggest authors to include or highlight the name of the inhibitors/modulators of SHP2.
what are the limitations of currently available inhibitors, clinical trials if any. How we can rectify them.
Reviewer 2 Report
The review presented by Saint-Laurent et al., titled “The tyrosine phosphatase SHP2, a new target for insulin 2 resistance?” is focused on the possible role of the tyrosine phosphatase SHP2 and its role in insulin resistance.
The text include a wide extension of SHP2 (from structure to function) demonstrating an extensive knowledge of the topic. However, some minor comments will be considered to improve the actual version:
- The localization of SHP2 is ubiquitous and the enzyme can be found in the cytosol, nucleus or mitochondria (lines 108-116), but it has been described if there is any situation that favors its location in some organelle or cytosol?. This information is not detailed in the text.
- Based on the title and the main objective of the review, the section 1.3. Pathological roles in cancer and genetic diseases should be removed.
- The section 2.2. A direct and dual role of SHP2 in insulin signaling is probably one of the most important topic described in the review but is too short and only describe effects on liver, however, the review include additional information from other tissues that should be used to extend the knowledge the dual role of SHP2 in other tissues (such as muscle, adipose tissue…) commented in lines 198-202 or resumed in table 1.
- The title of the figure 1 SHP2 in insulin resistance is confused due to explain the role of SHP2 in insulin resistance but include the 2 situations: insulin resistance (in red) or against insulin resistance (in green). Thus, the use of Dual role of SH2 in insulin signaling or SHP2 in insulin signaling should be more appropriate.
